evolution, immunology

sexual selection, genetic compatibility, cervical mucus, MHC, cryptic female choice, infertility

**Author for correspondence:**
Jukka Kekäläinen
e-mail: jukka.s.kekalainen@uef.fi

# Post-copulatory genetic matchmaking: HLA-dependent effects of cervical mucus on human sperm function

Annalaura Jokiniemi[1], Martina Magris[1], Jarmo Ritari[2], Liisa Kuusipalo[3], Tuulia Lundgren[1], Jukka Partanen[2] and Jukka Kekäläinen[1]

[1]Department of Environmental and Biological Sciences, University of Eastern Finland, PO Box 111, 80101 Joensuu, Finland
[2]Finnish Red Cross Blood Service, Research and Development, Haartmaninkatu 8, 00290 Helsinki, Finland
[3]North Karelia Central Hospital, Tikkamäentie 16, 80210 Joensuu, Finland

AJ, 0000-0001-9566-8385; MM, 0000-0003-3328-6343; JR, 0000-0002-3458-9314; JP, 0000-0001-6681-4734; JK, 0000-0001-6303-6797

Several studies have demonstrated that women show pre-copulatory mating preferences for human leucocyte antigen (HLA)-dissimilar men. A fascinating, yet unexplored, possibility is that the ultimate mating bias towards HLA-dissimilar partners could occur after copulation, at the gamete level. Here, we explored this possibility by investigating whether the selection towards HLA-dissimilar partners occurs in the cervical mucus. After combining sperm and cervical mucus from multiple males and females (full factorial design), we found that sperm performance (swimming velocity, hyperactivation, and viability) was strongly influenced by the male–female combination. This indicates that sperm fertilization capability may be dependent on the compatibility between cervical mucus (female) and sperm (male). We also found that sperm viability was associated with partners' HLA dissimilarity, indicating that cervical mucus may selectively facilitate later gamete fusion between immunogenetically compatible partners. Together, these results provide novel insights into the female-mediated sperm selection (cryptic female choice) in humans and indicate that processes occurring after copulation may contribute to the mating bias towards HLA-dissimilar partners. Finally, by showing that sperm performance in cervical mucus is influenced by partners' genetic compatibility, the present findings may promote a deeper understanding of infertility.

## 1. Introduction

Reproductive success is strongly dependent on the genetic compatibility of the reproductive partners [1–4]. Accordingly, individuals often differ in their mating preferences, ensuring that prospective offspring receive an optimal combination of parental genotypes [1,5,6]. Accumulating evidence suggests that in several species, genetic compatibility of the partners is largely determined by the major histocompatibility complex (MHC) immune genes. Supporting this view, many vertebrate species select MHC-dissimilar mating partners and this way strive to increase MHC diversity and immunocompetence of their offspring [7–11]. MHC disassortative mating preferences have been thought to occur predominantly prior to copulation [12–16] and especially MHC-associated odour signals have been assumed to play an important role in the process. However, MHC-based mate choice has been shown to also operate after copulation [17–23] via female-controlled paternity biasing processes (cryptic female choice [24–29]), which may ultimately determine, which parental MHC alleles are eventually transmitted to offspring. Nevertheless, the mechanistic basis of the cryptic female choice has remained largely unclear, especially in mammals

and other internally fertilizing species (but see [30], for recent evidence for cryptic female choice in humans).

Along with numerous animal taxa, MHC diversity has also been reported to be advantageous for immunocompetence and reproductive success in humans [31–35]. Accordingly, a number of studies have demonstrated that both women and men show pre-copulatory mating preferences for the body odours of HLA (human leucocyte antigen)-dissimilar partners [36–39]. Interestingly, recent studies have shown that both HLA molecules and HLA-linked olfactory receptors that detect these molecules are present on the surface of human sperm ([40–43], although HLA expression on mature sperm remains controversial [44,45]). Furthermore, in addition to these sperm surface molecules, soluble HLAs are found in the secretions of the female reproductive tract [46,47]. Together, these findings indicate that the definitive mate choice for paternal HLA genes could occur at the gamete level. Indeed, accumulating evidence shows that female-derived reproductive secretions can provide multiple opportunities for gamete-mediated mate choice, potentially favouring the fusion of immunologically compatible gametes (reviewed in [48]). However, MHC- (or HLA-) associated gamete-level mating preferences remain to be experimentally demonstrated in mammals (but see [18,21]).

In humans, only about one of every million spermatozoa is able to pass the lower reproductive tract of women and thus reach the oviducts [49]. One of the first barriers encountered by human sperm on their way to the egg is the cervical mucus. Cervical mucus is a viscoelastic non-Newtonian fluid secreted by the cervical glands [50] and is primarily comprised of mucins (large glycoproteins) that form a thick fibrous network [51]. Prior to ovulation, the viscosity and elasticity of cervical mucus decreases significantly, allowing a specific sperm sub-population to penetrate into the uterus (e.g. [50]). Accordingly, cervical mucus has been thought to play an important role both in sperm function regulation and in sperm transport through the cervix. Thus, cervical mucus may simultaneously aid migration of functionally normal sperm and serve as a selective barrier for abnormal and poorly motile sperm cells [52]. Cervical mucus is also involved in the initiation of sperm capacitation [53] and stimulation of hyperactivation [54], and it has been shown to maintain sperm function by preventing the spontaneous (premature) acrosome reaction [53,55]. Given that hyperactivated motility is believed to be advantageous especially after the sperm have reached the oviduct [56], and since premature hyperactivation may be detrimental for sperm fertilization capability [57–59], the functional significance of sperm hyperactivation in the cervical mucus is unclear. Accordingly, the mechanisms and criteria of cervical mucus-induced sperm selection also remained unknown.

Cervical mucus contains a wide array of immunological molecules [50,60], including soluble HLAs. Following insemination, the cervix is the site of a conspicuous inflammatory response, involving an influx of inflammatory cells and a change in cytokine, chemokine, and immunoglobulin gene expression [61,62]. Interestingly, the strength of this post-copulatory female immune response shows considerable variation across individual males [63], which may potentially lead to non-random sperm selection via selective phagocytosis of superfluous (and possible genetically incompatible: see [64]) sperm cells [63,65]. On the other hand, it has been hypothesized that along with selective elimination of sperm cells, cervical mucus may also promote the storage of specific sub-populations of sperm for later use in fertilization [62,66]. Collectively, these findings indicate that cervical mucus may have an important role in facilitating sperm selection towards immunologically compatible partners.

In the present study, we explored this novel possibility by investigating whether the chemical composition of human cervical mucus could mediate sperm selection towards HLA-dissimilar males. To separate these chemically mediated effects from the physical (structural) effects of the cervical mucus, we first ruptured the mucin network of the mucus samples and diluted the samples to lower their viscosity. Then, by using a full factorial (North Carolina II) design, we treated the sperm of eight men with the cervical mucus of nine women in all possible combinations ($N = 72$) and measured sperm motility (i.e. swimming velocity and hyperactivation) and viability in each male–female combination. We genotyped all the individuals by a genome-wide single nucleotide polymorphism (SNP) array and imputed their HLA class I and II alleles at the unique protein sequence level. Finally, we tested whether male–female HLA similarity (or genome-wide similarity) predicts sperm performance and survival in cervical mucus. We hypothesized that sperm selection for genetic (HLA) compatibility is influenced by cervical mucus and favours the sperm of HLA-dissimilar males.

## 2. Material and methods

### (a) Study subjects and sample collection

Female participants ($N = 9$, mean age 33.3 ± 1.7 s.e. years) were recruited from the fertility clinic of the North Karelia Central Hospital, Finland. Eight of the females were Caucasian and one them was Asian. All the participants were undergoing artificial insemination or transvaginal follicular aspiration for *in vitro* fertilization; cervical mucus samples were collected in conjunction with these procedures. All female subjects were hormonally treated to stimulate ovulation and cervical mucus samples were collected during women's fertile days and after at least four days from the last unprotected intercourse. Samples were collected with a Pipelle de Cornier® Mark II catheter (Laboratoire CCD, Paris, France). In order to rupture the mucin network and to lower the viscosity of the mucus (and thus to investigate the role of cervical mucus' chemical composition in sperm selection), all the samples were diluted 1 : 6 (weight: volume) with PureSperm® Wash solution (Nidacon International AB, Mölndal, Sweden), vortexed, and finally sonicated. All the samples were kept at 4°C overnight and then centrifuged 500 × g for 10 min. The supernatant was separated and divided into 60 µl aliquots and stored in liquid nitrogen until later use. Before the dilution and sonication, two independent aliquots from each mucus sample were microscopically examined (400 × magnification) to ensure that no spermatozoon was present in the samples.

Male participants ($N = 8$) were volunteers recruited via the same fertility clinic (see above). All the participants were Caucasian, and their mean age was 33.4 (±1.6 s.e.) years. All the males were normozoospermic according to World Health Organization (WHO) criteria [67]. All the participants provided semen samples by masturbation after 1 to 7 days of sexual abstinence. After 30 min of liquefaction (37°C), the ejaculate was washed with density gradient centrifugation according to the manufacturer's protocol (with PureSperm® 40 and 80 gradients, Nidacon). Finally, sperm pellets were resuspended in PureSperm® Wash solution (Nidacon) to the final concentration of 42 (±1.4 s.e.) million sperm ml$^{-1}$. An informed consent from all the subjects (both females and males) was obtained by Liisa Kuusipalo.

## (b) Treatment of sperm with cervical mucus

Diluted cervical mucus (CM) samples of each of the nine women were thawed and divided in two independent sub-samples (A and B: 18 samples in total). Washed sperm aliquots from each of the eight men were then combined (1:1) with each of these CM sub-samples (27 µl sperm + 27 µl diluted cervical mucus) in all possible male–female combinations (hereafter CM treatment). This full-factorial design yielded 144 male–female combinations in total (8 males × 9 females × 2 sub-samples). All the samples were kept at 37°C during the whole experimental period. For each male, all the sperm measurements (see below) were performed on the day of semen collection (i.e. by using fresh sperm).

## (c) Sperm motility and viability measurements

Sperm motility measurements were performed by adding 1 µl of each CM-treated sperm sample to pre-warmed (+37°C) Leja 4-chamber (chamber height 20 µm) microscope slides (Leja, Nieuw-Vennep, the Netherlands). Then, the effect of CM treatment on sperm motility (curvilinear velocity: VCL; linearity of the swimming trajectory: LIN; and amplitude of the lateral head displacement: ALH) was measured using computer-assisted sperm analysis (CASA; Integrated Semen Analysis System, ISAS v. 1.2 Proiser, Valencia, Spain), with a negative phase-contrast microscope (100 × magnification) and a capture rate of 100 frames $s^{-1}$. Furthermore, following Kay & Robertson [68], the hyperactivated state of the sperm was determined based on following three CASA parameters: VCL > 150 µm $s^{-1}$, LIN < 50%, and ALH > 2.0. Sperm motility was recorded at three time points: one, three, and five hours since the beginning of the CM treatment. Sperm motility was measured on average from $680 \pm 11$ sperm (mean ± SE) per male–female combination.

At the end of the motility measurements, a 25 µl aliquot from each CM-treated sperm sample was stained with propidium iodide (PI, 5 µg $ml^{-1}$). After three minutes of incubation (in the dark), 0.5 µl of 1% formalin was added to immobilize the sperm. Then, 10 µl of PI-treated sperm samples were individually transferred to a LUNA™ Reusable Slide (Logos Biosystems, Annandale, VA, USA) and the number of dead and total sperm cells were measured using a LUNA-FL™ Dual Fluorescence Cell Counter (Logos Biosystems). Viability was calculated as the proportion of live spermatozoa (not PI-stained) on the total number of counted spermatozoa. Sperm viability was measured on average from $44\,190 \pm 1\,855$ sperm (mean ± SE) per male–female combination.

All the sperm motility and viability measurements (see above) included two independent replicate recordings within both sub-samples, in each of the 72 male–female combinations, resulting in 288 (72 combinations × 2 sub-samples × 2 replicates) recordings in total. To minimize a potential time effect on the measured sperm traits, both the initiation of CM treatment and subsequent sperm motility and viability measurements in the first sub-sample (A) were always conducted in the following order (with 3 min intervals): CM1, CM2, … , CM9, whereas in the second sub-sample (B) initiation of treatment and sperm measurements were performed in the opposite order: CM9, CM8, … , CM1. This time delay accounts for the time needed for sperm motility and sperm viability measurements; this way, each sub-sample was measured after an identical interval from the beginning of the CM treatment.

## (d) Genotyping of the study subjects

DNA of all the 17 subjects was extracted from EDTA blood according to the manufacturer's instructions using a PureLink® Genomic DNA Kit (Invitrogen). Extracted DNA samples were genotyped on an Illumina Global Screening Array-24 v2.0 kit at the Institute for Molecular Medicine Finland (FIMM). HLA imputation of

seven classical HLA genes (HLA-A, -B, -C, -DRB1,- DQA1, -DQB1, and -DPB1) at four-digit (i.e. protein level) resolution was performed with R version 3.4.4, using the package HIBAG, version 1.22.0 [69] with default settings and the European and Asian reference panels (European-HLA4-hg19.Rdata and Asian-HLA4-hg19.Rdata, respectively) for the human genome build GRCh37/hg19. HLA similarity in all male–female combinations was determined by: (1) calculating the number of shared HLA alleles (0–14) over the seven imputed HLA genes, (2) calculating the Grantham's distance [70] between male and female HLA alleles. The Grantham distance accounts for the functional properties of HLA molecules as it considers the molecular volume of amino acids composing HLA alleles, which define the peptides that will bind to the HLA molecule. The Grantham distance calculations were based on HLA allele amino acid sequence alignments downloaded from the Immuno Polymorphism Database (IPD)-International ImMunoGeneTics project (IMGT)/HLA database [71] release 3.38.0 (ftp://ftp.ebi.ac.uk/pub/databases/ipd/imgt/hla/alignments/). The whole-genome distance (genome-wide similarity) measure in all male–female combinations was calculated as the total number of shared genotypes for all genotyped and quality filtered bi-allelic SNPs.

## (e) Statistical analyses

Repeatability of sperm motility parameters (within each time point) and sperm viability measurements were calculated between sub-samples and between replicate measurements within sub-samples according to [72]. The effect of male, female, and male–female interaction (combination) on sperm swimming velocity (VCL), hyperactivation (percentage of hyperactivated sperm cells), and sperm viability were tested in linear mixed-effects models (LMM). In order to account for repeated measures of VCL and hyperactivation over time, the full models for these sperm traits included random slope of time (i.e. time point) for sub-samples (time point | sub-sample), sub-sample-replicate interaction (time point | sub-sample:replicate), male (time point | male), female (time point | female), and male–female interaction (time point | male:female) [73]. Furthermore, time point (60, 180, and 300 min) was included as a fixed factor in the models. The associations between male–female HLA similarity or genome-wide similarity and measured sperm traits were modelled by adding (a) the number of shared HLA alleles, (b) the Grantham distance of HLA alleles, or c) genome-wide SNP similarity in each ($N = 72$) male–female combination as an additional fixed effect (covariate) in the above-mentioned models. Initial models also included the interaction between time point and genetic parameter. Models with genetic parameters were simplified (based on Akaike information criterion (AIC)) by replacing random slope (time) of sub-sample-replicate interaction with random intercept (1 | sub-sample:replicate) in case the full model did not converge (electronic supplementary material, datafile S1). Furthermore, models were simplified by removing the interaction between time point and above-mentioned genetic parameters if these interactions did not improve model fit (based on AIC values). As the effect (slope) of time point was significant (see results) in the models, we also ran time point-specific models for both VCL and hyperactivation. Time point-specific analyses were also performed when the slope of the genetic parameters was dependent on the time point (i.e. when a statistically significant genetic parameter × time point interaction was found). Given that sperm viability was determined only at the end of the motility measurements, time point-specific analyses were not required for sperm viability.

Time point-specific models for VCL and hyperactivation (and sperm viability models) included CM treatment sub-sample, the interaction between sub-sample and replicate, male effect, female effect, and male–female interaction effect, which were all modelled as random factors. Furthermore, as described above,

**Table 1.** Overall linear mixed model (LMM) statistics for sperm swimming velocity (VCL) and hyperactivation in the cervical mucus. LMM included time point as a fixed effect and the random slope of time point for males, females, male–female interaction, sub-samples, and sub-sample-replicate interaction (random effects).

| effects | VCL | | | hyperactivation | | |
|---|---|---|---|---|---|---|
| random | $\chi^2$ | d.f. | *p*-value | $\chi^2$ | d.f. | *p*-value |
| time point \| male | 53.67 | 5 | **<0.001** | 53.57 | 5 | **<0.001** |
| time point \| female | 44.57 | 5 | **<0.001** | 35.20 | 5 | **<0.001** |
| time point \| male × female | 235.90 | 5 | **<0.001** | 208.42 | 5 | **<0.001** |
| time point \| sub-sample | 4.296 | 5 | 0.508 | 4.01 | 5 | 0.547 |
| time point \| sub-sample × replicate | 0.005 | 5 | 1 | 0.22 | 5 | 0.999 |
| **fixed** | ***t*** | **d.f.** | ***p*-value** | ***t*** | **d.f.** | ***p*-value** |
| intercept | 14.22 | 14 | **<0.001** | 5.36 | 12 | **<0.001** |
| time point 60 versus 180 | −5.27 | 12 | **<0.001** | −4.76 | 13 | **<0.001** |
| time point 60 versus 300 | −4.95 | 13 | **<0.001** | −4.30 | 12 | **0.001** |

these models were used to study the association between three genetic parameters (fixed factor) and measured sperm traits. In order to investigate whether the slope of the genetic parameters differs across males, we also modelled the interaction between genetic parameter (fixed factor) and male (random factor) as additional random factors. Models were simplified by removing interactions between male and genetic parameters if these interactions did not improve model fit (based on AIC values).

Finally, we calculated the relative proportion of total variance explained by individual random factors. Model assumptions were graphically verified using Q-Q plots and residual plots. All *p*-values presented are from two-tailed tests, with $\alpha = 0.05$. The statistical analyses were conducted using the packages lmerTest in R (v. 3.4.4) [74].

## 3. Results

### (a) Sperm motility and viability

Sperm swimming velocity (VCL) and the proportion of hyperactivated sperm cells were affected by time point and the effect (slope) of time differed across males, females, and male–female combinations (table 1). Time point-specific analyses revealed that both VCL (table 2) and hyperactivation (table 3 and figure 1) were affected by male and male–female interaction at all three time points, while the female effect was significant only for the first two time points (60 and 180 min). The magnitude of male–female interaction effect on VCL and hyperactivation increased with time, explaining 24.3–32.3% of total variance in sperm motility in the last two time points (180 and 300 min; tables 2 and 3). Sperm viability was affected by male and male–female interaction, whereas the female-effect was not significant (table 4 and figure 2). Male–female interaction explained 8.5% of the total variation in sperm viability. All three sperm parameters (VCL, hyperactivation, and viability) were highly repeatable both between replicate measurements within sub-samples ($R > 0.81$; $p \ll 0.001$) and between sub-samples ($R > 0.70$; $p \ll 0.001$).

### (b) The effect of HLA similarity and genome-wide similarity on sperm performance

Overall models (including all time points) showed that HLA allele sharing and genome-wide similarity of males and females did not influence VCL (number of different HLA alleles: $F_{1,64} = 0.13$, $p = 0.72$; genome-wide similarity: $F_{1,11} = 0.79$, $p = 0.39$) or hyperactivation (number of different HLA alleles: $F_{1,65} = 0.02$, $p = 0.90$; genome-wide similarity: $F_{1,11} = 0.81$, $p = 0.39$). Conversely, VCL and hyperactivation were influenced by the Grantham distance (VCL: $F_{1,64} = 4.08$, $p = 0.047$; hyperactivation: $F_{1,64} = 4.76$, $p = 0.033$). The interaction between the Grantham distance and time point was significant for both VCL and hyperactivation (VCL: $F_{2,65} = 4.97$, $p = 0.010$; hyperactivation: $F_{2,65} = 4.99$, $p = 0.010$). In other words, the effect of the Grantham distance on sperm motility differed across time points. Time point-specific models showed that VCL and hyperactivation were negatively affected by Grantham distance, but only at 180 min (VCL: $t_{13} = -2.78$, $p = 0.016$; hyperactivation: $t_{11} = -2.42$, $p = 0.034$). Furthermore, the effect of the Grantham distance varied across different males (the Grantham distance × male interaction) for both sperm traits (VCL: $\chi^2 = 10.85$, $p = 0.004$; hyperactivation: $\chi^2 = 14.37$, $p < 0.001$).

Sperm viability was significantly (positively) affected by the number of different HLA alleles (figure 3) and by the Grantham distance (i.e. HLA dissimilarity), but not by genome-wide genetic distance (number of different HLA alleles: $t_{63} = 3.94$, $p < 0.001$; the Grantham distance: $t_{52} = 2.87$, $p = 0.006$; genome-wide distance: $t_7 = 0.46$, $p = 0.661$). The interaction between HLA dissimilarity and male was not statistically significant, indicating that the effect of HLA dissimilarity on sperm viability was similar across the eight males (number of different HLA alleles: $\chi^2 = 4.64$, $p = 0.099$; the Grantham distance: $\chi^2 = 5.26$, $p = 0.072$). Together, these results show that sperm had a higher probability of survival in HLA-dissimilar male–female combinations compared to more similar combinations.

## 4. Discussion

Our results show that along with sperm intrinsic quality (male effect) and cervical mucus identity (female effect), sperm performance was also strongly dependent on male–female combination (interaction), explaining 8.5–32.3% of the total variation in measured sperm traits. In other words, females had a stronger effect on the sperm of some males than the others. We also observed that both the number of different HLA alleles and the Grantham pairwise

**Table 2.** Linear mixed model statistics for the effect of male, female, and male–female interaction (random effects) on sperm swimming velocity (VCL) at three time points (60, 180, and 300 min) after the initiation of the cervical mucus treatment. The models also included sub-sample and the interaction between sub-sample and replicate as additional random effects. % var = proportion of total variance explained by random factors.

| effects | 60 min | | | | 180 min | | | | 300 min | | | |
|---|---|---|---|---|---|---|---|---|---|---|---|---|
| random | $\chi^2$ | d.f. | p-value | % var | $\chi^2$ | d.f. | p-value | % var | $\chi^2$ | d.f. | p-value | % var |
| male | 63.15 | 1 | **<0.001** | 47.00 | 34.23 | 1 | **<0.001** | 40.47 | 47.11 | 1 | **<0.001** | 54.16 |
| female | 45.66 | 1 | **<0.001** | 30.23 | 10.68 | 1 | **0.001** | 13.30 | 0.83 | 1 | 0.364 | 2.12 |
| male × female | 137.02 | 1 | **<0.001** | 14.17 | 112.47 | 1 | **<0.001** | 27.21 | 92.85 | 1 | **<0.001** | 24.34 |
| sub-sample | 2.40 | 1 | 0.122 | 1.20 | 2.21 | 1 | 0.137 | 1.38 | 0 | 1 | 1 | 0 |
| sub-sample × replicate | 0.42 | 1 | 0.516 | 0.08 | 0 | 1 | 1 | 0 | 0.44 | 1 | 0.505 | 0.17 |
| residual | | | | 7.32 | | | | 17.64 | | | | 19.21 |
| fixed | t | d.f. | p-value | | t | d.f. | p-value | | t | d.f. | p-value | |
| intercept | 14.23 | 13.9 | **<0.001** | | 15.13 | 11.0 | **<0.001** | | 17.86 | 7.5 | **<0.001** | |

**Table 3.** Linear mixed model statistics for the effect of male, female, and male–female interaction (random effects) on the proportion of hyperactivated sperm cells at three time points (60, 180, and 300 min) after the initiation of the cervical mucus treatment. The models also included sub-sample and replicate as additional random effects. % var = proportion of total variance explained by random factors.

| effects | 60 min | | | | 180 min | | | | 300 min | | | |
|---|---|---|---|---|---|---|---|---|---|---|---|---|
| random | $\chi^2$ | d.f. | p-value | % var | $\chi^2$ | d.f. | p-value | % var | $\chi^2$ | d.f. | p-value | % var |
| male | 62.96 | 1 | **<0.001** | 49.69 | 20.51 | 1 | **<0.001** | 27.61 | 31.97 | 1 | **<0.001** | 43.85 |
| female | 35.18 | 1 | **<0.001** | 23.14 | 8.07 | 1 | **0.004** | 12.59 | 0.78 | 1 | 0.377 | 2.65 |
| male × female | 99.88 | 1 | **<0.001** | 14.37 | 86.84 | 1 | **<0.001** | 31.02 | 111.81 | 1 | **<0.001** | 32.32 |
| sub-sample | 2.67 | 1 | 0.102 | 2.16 | 1.58 | 1 | 0.209 | 2.34 | 0 | 1 | 1 | 0 |
| sub-sample × replicate | 0.49 | 1 | 0.483 | 0.13 | 0.39 | 1 | 0.530 | 0.28 | 0.33 | 1 | 0.565 | 0.16 |
| residual | | | | 10.52 | | | | 26.17 | | | | 21.02 |
| fixed | t | d.f. | p-value | | t | d.f. | p-value | | t | d.f. | p-value | |
| intercept | 5.36 | 12.5 | **<0.001** | | 3.43 | 9.7 | **0.007** | | 3.06 | 7.7 | **0.017** | |

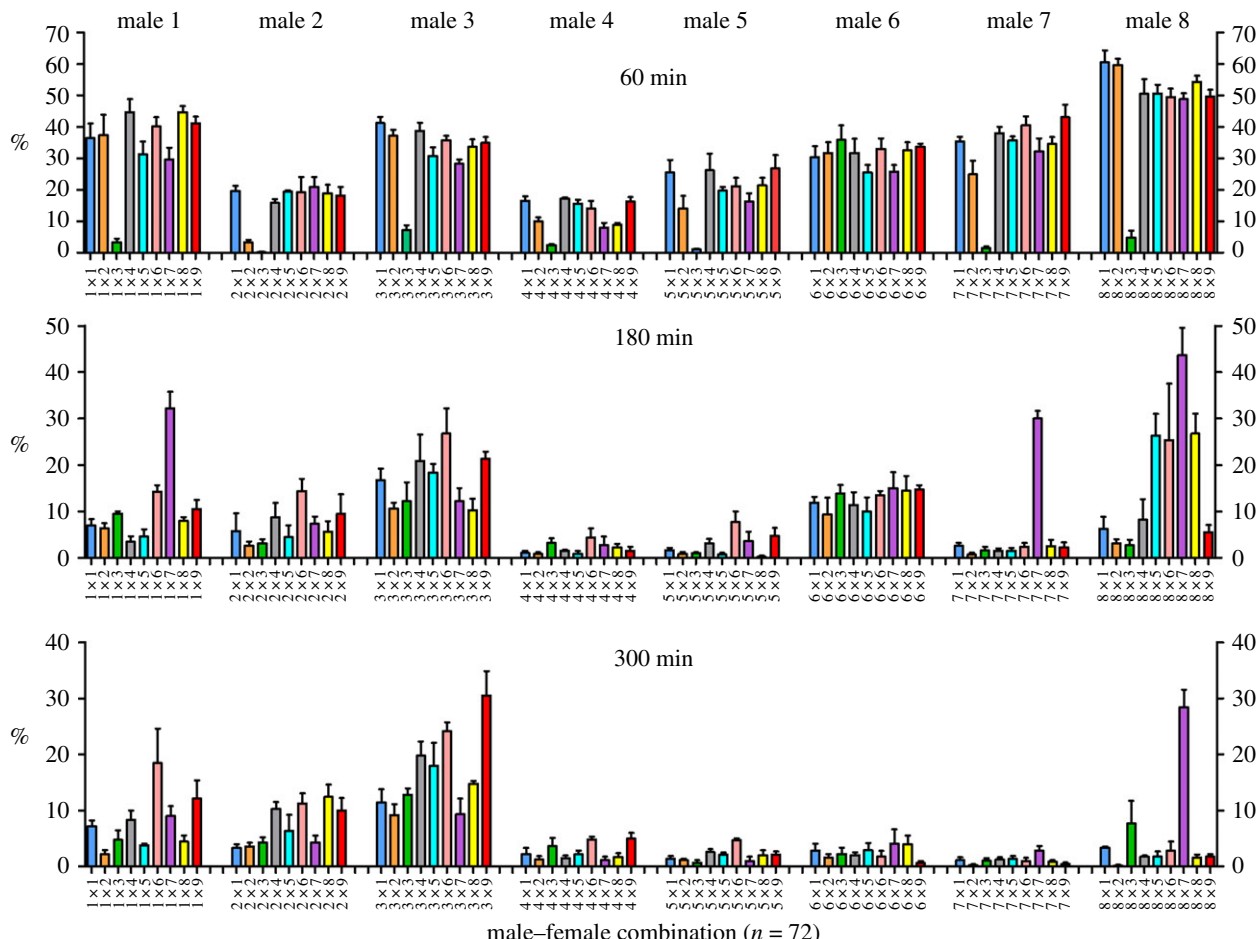

**Figure 1.** The effect of male–female interaction (combination) on the proportion of hyperactivated sperm cells (calculated on four replicate measurements per combination ± s.e.) at three time points (60, 180, and 300 min) after the initiation of the cervical mucus treatment. Bar colours represent female identity (*N* = 9).

**Table 4.** Linear mixed model statistics for the effect of male, female, and male–female interaction (random effects) on the proportion of live sperm in the cervical mucus. The models also included sub-sample and the interaction between sub-sample and replicate as additional random effects. % var = proportion of total variance explained by random factors.

| effects | | | | |
|---|---|---|---|---|
| random | $\chi^2$ | d.f. | *p*-value | % var |
| male | 104.03 | 1 | **<0.001** | 76.73 |
| female | 0.46 | 1 | 0.500 | 0.62 |
| male × female | 42.06 | 1 | **<0.001** | 8.54 |
| sub-sample | 0.47 | 1 | 0.491 | 0.25 |
| sub-sample × replicate | 0.04 | 1 | 0.838 | 0.04 |
| residual | | | | 13.81 |
| fixed | *t* | d.f. | *p*-value | % var |
| intercept | 33.26 | 7.3 | **<0.001** | |

amino acid distance of HLA alleles affected sperm viability: sperm had higher survival rates in HLA-dissimilar male–female combinations than in more similar combinations. Conversely, we found that HLA-allele sharing did not affect sperm motility, but the Grantham distance was negatively associated with sperm swimming velocity and hyperactivation in one of the three time points (180 min). In other

words, sperm motility was higher in male–female combinations that had low HLA amino acidic divergence in comparison to more dissimilar combinations. However, the observed effect was male-dependent, indicating that the effect of the Grantham distance on sperm motility varies across males. None of the measured sperm traits was associated with male–female genome-wide similarity. Although all the male subjects were diagnosed as normozoospermic it is possible that reproductive physiology of the study subjects may partly differ from that of the average male and/or female population. Consequently, some caution should be applied to generalize our findings and future research should ideally aim to test whether the same mechanisms are widespread in the human population.

The female reproductive tract allows only a minute subset of spermatozoa to reach the site of fertilization [49,75,76], but the mechanisms and function of this stringent sperm selection have remained ambiguous. It has been suggested that the functional incompatibility between cervical mucus and sperm could play an important role in the process [62,66], but to the best of our knowledge none of the earlier studies have experimentally tested this possibility. The present results demonstrate that the chemical composition of the cervical mucus may selectively maintain sperm viability of HLA-dissimilar males, indicating that cervical mucus could mediate post-copulatory choice towards the sperm of immunologically compatible males. Furthermore, given that sperm performance in cervical mucus predicts fertilization success [62], these

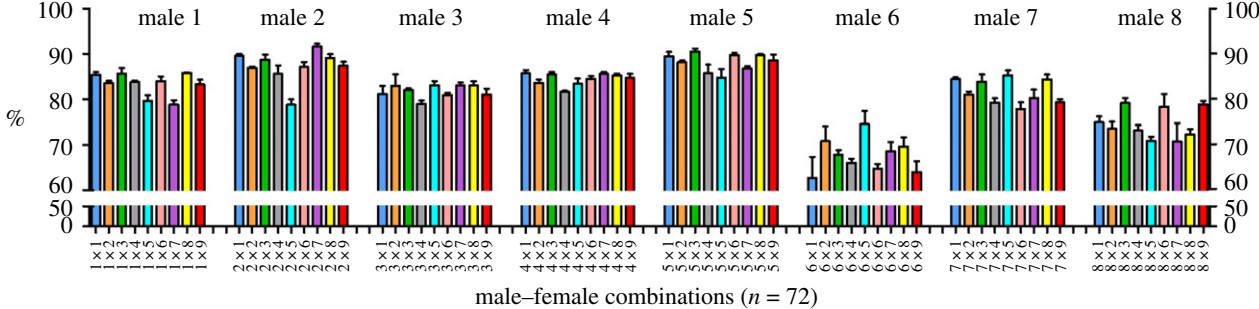

**Figure 2.** The effect of male–female interaction (combination) on the proportion of live sperm cells (calculated on four replicate measurements per combination ± s.e.) in the cervical mucus. Bar colours represent female identity ($N = 9$).

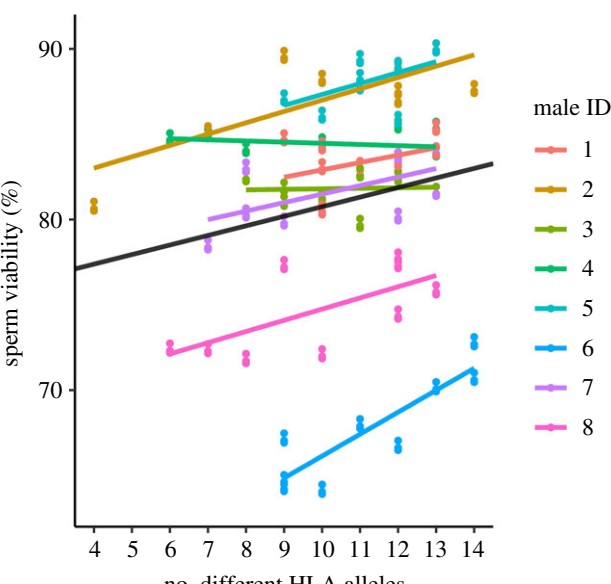

**Figure 3.** Sperm viability (proportion of live cells) was positively associated with male–female HLA dissimilarity. Datapoints represent fitted values obtained from the LMM. Male-specific associations are identified by different colours and the black line represents the average slope across all males. For each male–female combination, viability was determined based on two replicate measurements from each of the two sub-samples ($n = 4$ measurements in total, see 'Material and Methods', for more details). The slope of the associations did not vary across males ($p = 0.099$).

results raise the novel possibility that immunological compatibility between sperm and cervical mucus plays an important role in determining the reproductive success of the partners. While previous studies have reported an effect of MHC-dissimilarity on egg-sperm fusion [6,17–19,21], to our knowledge, this is the first study to show that MHC-based cryptic female choice could be mediated by fluids of the female reproductive tract in mammals.

Earlier studies have demonstrated that cervical mucus is capable of conserving sperm function [62] and it has been hypothesized that cervical crypts could serve as sperm reservoirs [77], where sperm motility is restrained to enhance longevity, such as in the sperm storage sites of the oviduct [78,79]. Besides demonstrating that cervical mucus likely preserves sperm viability of HLA-dissimilar males, we also found evidence that cervical mucus may simultaneously restrict sperm motility of these males. This could indicate that one key function of cervical mucus is to selectively store the sperm of immunologically compatible males, possibly for later use in fertilization. Alternatively, our results raise the intriguing possibility that females (via cervical mucus) may reduce

the survival of sperm of (overly) HLA-similar males on the one hand and slow down the sperm of (overly) HLA-dissimilar ones on the other. Such a search for intermediate MHC dissimilarity may facilitate 'production' of intermediately heterozygous offspring at MHC loci, which, according to the optimal MHC-heterozygosity hypothesis, may have better immunocompetence than more heterozygous individuals [80]. In fact, while highly heterozygous individuals at MHC loci are able to present more antigens to the immune system, they are likely to have smaller T-cell repertoires following thymic selection [80]. Supporting the optimal heterozygosity hypothesis, Jacob *et al.* [81] showed that women prefer the body odour of men with whom they share few HLA alleles over the more similar and dissimilar men [36]. However, since observed sperm motility associations were present only at one time point and were not consistent across males, further studies are required to investigate the relative importance of these alternative functions of cervical mucus. Furthermore, since performed cervical mucus dilution likely altered sperm motility patterns, we encourage future studies to confirm whether our results can be replicated in undiluted cervical mucus to fully account for its natural viscoelastic features.

The observed effects of partners' HLA dissimilarity on sperm function could potentially arise at least through two evolutionary mechanisms. First, as highlighted above, cervical mucus-mediated selection towards HLA-dissimilar males may represent an evolutionary strategy that ensures the 'production' of offspring that have broad (or optimal) antigen recognition capability and thus better ability to fight against infections [82]. Supporting this possibility, pathogens have been widely assumed to be the strongest selective agent in human evolution (e.g. [9]). Alternatively, it is possible that HLA-associated sperm preferences represent a gamete-level inbreeding avoidance mechanism that prevents mating between close relatives [10]. In the present study, however, we found that overall (genome-wide) genetic similarity was not associated with sperm performance, indicating that observed findings are more likely to be a direct consequence of HLA-dependent sperm selection, instead of inbreeding avoidance based on HLA-independent cues. Although detailed molecular-level mechanisms behind our findings remain to be described, we envisage that sperm surface HLA molecules and/or HLA-linked olfactory receptors [83] may play an important role in the demonstrated sperm selection process.

Besides clarifying the mechanistic basis of female-induced sperm selection in humans, our findings may have novel implications for the deeper understanding of infertility and for the development of new contraceptives. Infertility affects about 15% of couples globally and currently remains unexplained

in 30–40% of cases [84]. Furthermore, a reliable diagnosis of infertility is extremely challenging, and the accuracy of diagnoses is low when compared to several other areas of medicine [85,86]. According to current clinical practice, infertility problems are partitioned into male- and female-derived pathological factors and are thus thought to represent a disease of the reproductive system [87]. Our results indicate that this may represent an overly simplistic view, since it does not consider the fact that some male–female (or their gamete) combinations may be immunologically more compatible than others. Consequently, gamete-level incompatibility may reduce the probability of conception and may help to understand fertilization problems, especially in couples that are diagnosed with unexplained infertility.

In conclusion, our results show that chemical factors in the cervical mucus preferentially conserve sperm viability (and possibly constrain sperm motility) of HLA-dissimilar males. This indicates that one of the key functions of cervical mucus may be to selectively facilitate gamete fusion between immunogenetically compatible partners and this way facilitate optimization of offspring immunocompetence. Immunological mechanisms of sperm selection have remained virtually unexplored in mammals and internally fertilizing species, in general. The present results provide novel insights into

MHC-based post-copulatory sperm selection in humans and may be potentially applied to many other species. Furthermore, a more pervasive integration of the demonstrated 'gamete compatibility' concept into current infertility diagnostic guidelines may facilitate development of more personalized infertility diagnostics and increase accuracy of the diagnoses.

Data accessibility. Due to the privacy commitments of the patients and the terms of the consent, the original data cannot be made widely available to the academic community.

Authors' contributions. J.K. designed and oversaw the study, A.J, T.L., and L.K. performed the experiments, M.M., J.R., and J.K. analysed the data, A.J., M.M., J.K., J.R., and J.P. wrote the paper. All authors read and approved the final version of the manuscript.

Competing interests. The authors declare no conflict of interest.

Funding. This study was financially supported by the Academy of Finland (grant no. 308485 to J.K., grant no. 314105 to M.M and A.J. and grant no. 288393 to J.R. and J.P.) and by the VTR funding 314105 by the Finnish Government (to J.R. and J.P.).

Acknowledgements. We want to thank staff of the fertility clinic of the North Karelia Central Hospital and ISLAB for sample collection, Kati Donner (Institute for Molecular Medicine Finland) for genotyping the samples, and Lauri Mehtätalo for statistical consultation. The sample collection and the experimental procedures were based on the favourable opinion of the Ethics Committee, Hospital District of Northern Savo, Finland (decision number 77/2017).

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
