## [Reviewer comments · Proceedings of the Royal Society B: Biological Sciences]

Review History

RSPB-2020-1060.R0 (Original submission)

Review form: Reviewer 1

Recommendation

Accept with minor revision (please list in comments)

Scientific importance: Is the manuscript an original and important contribution to its field?

Excellent

General interest: Is the paper of sufficient general interest?

Good

Quality of the paper: Is the overall quality of the paper suitable?

Good

Is the length of the paper justified?

Yes

Should the paper be seen by a specialist statistical reviewer?

No

Do you have any concerns about statistical analyses in this paper? If so, please specify them explicitly in your report.

No

It is a condition of publication that authors make their supporting data, code and materials available - either as supplementary material or hosted in an external repository. Please rate, if applicable, the supporting data on the following criteria.

Is it accessible?

No

Is it clear?

N/A

Is it adequate?

N/A

Do you have any ethical concerns with this paper?

No

Comments to the Author

This manuscript investigates the physiological role and significance of cervical mucus-mediated effects on survival and function of sperm. The possibility of between individual female variance, between male variance, and interactions between male-female interactions were investigated in a full factorial design using clinical materials from 9 females and 8 males, and evaluated by linear mixed-model analysis. Compelling evidence of individual female-specific effects of cervical mucus on sperm motility, hyperactivation and viability were shown, in a male-partner specific manner. Additionally, sequencing analysis showed a clear relationship with extent of HLA disparity between male and female partners, indicating a female HLA-mediated mechanism.

This is an exciting and novel study that identifies a new mechanism by which post-copulatory cryptic female choice might operate in humans. The finding is important as, if a comparable process occurred in vivo, it reasonably would contribute to selective mechanisms impacting the likelihood of conception in a partner-specific manner. The study is well-designed, the analysis appears appropriate, and in general, the conclusions drawn are justified by the data provided. The paper is clearly and succinctly written, and relevant previous works are appropriately cited.

There are some issues listed below that if addressed, would improve the quality of the manuscript.

1. Abstract, li 24: "these results...indicate that the definitive mating bias towards HLA-dissimilar partners may occur after copulation". This sentence should be revised to recognise that other potential mechanisms for HLA-mediated mating bias may occur (these include pre-coital mechanisms such as pheromone-based partner selection, and perhaps other post-coital mechanisms operating at the level of embryo implantation). Consider "...indicate that one mechanism of bias towards HLA-dissimilar partners may occur after copulation".
2. Abstract, li 26 and discussion, li 350-351: "present findings raise the intriguing possibility that infertility may not represent exclusively a pathological condition". I recommend removing this statement from both the abstract and the discussion. Although I agree that here are likely biological mechanisms for increasing the prospects of successful mating depending on partner compatibility, and the current data indicate a new mechanism that could contribute to this, the study does not provide any evidence that might be construed to mean the selective mechanism is of sufficient force to achieve complete infertility. In fact, the data do not even show that selection occurs in vivo, so it is a very long bow to draw to impute a causal role in infertility. Moreover, the issue of infertility as pathology is a fraught and complex debate, and I do not agree

that the current data add to the evidence either way.

3. Introduction, li 51: The authors state that HLA molecules are present on sperm and give a reference (Sereshki et al. 2019). The cited paper shows FACS data indicating less than 20% of sperm express HLAI or HLAI, and FACS plots are not particularly convincing. There has been considerable debate on this topic for decades, with several studies supporting both points of view. It should be clear in the authors comments that there is data both ways and the issue remains contentious.

4. Introduction, li 75: "cervical mucus contains soluble HLA and their antibodies". It is my understanding that anti-HLA antibodies are only rarely detected in female tract secretions, and generally in the context of infertility. The authors should qualify this statement accordingly.

5. Introduction, li 96: "We hypothesised that sperm selection.... occurs in the cervix". This aspect of the hypothesis was not tested, as no in vivo experiments were conducted. Could the hypothesis be rephased to read "is influenced by cervical mucus", as this is what was tested?

6. Methods, li 119: What was the minimum number of days of abstinence? This should be added.

7. Methods li 127: Did the authors consider repeating the experiment with the same biological materials on more than one occasion, to evaluate between experiment variance? Did they consider testing the consistency of the male-female relationships over time, by collecting semen and/or mucus from the same individuals on more than one occasion? This would improve confidence in the reliability of the findings.

8. Discussion, li 339: The authors comment on possible mechanisms by which sperm-specific effects might be mediated, and imply that sperm-borne HLA and/or anti-HLA antibodies could contribute. Is this possible if only a very small proportion of sperm express HLAI and/or HLAI (see above), or if such antibodies are only present in a subset of women? Did they consider measuring these in their cervical mucus samples and relating the presence of antibodies to the male-specific effects? An additional comment on the limitations of this speculation would be warranted.

9. Discussion, li 321 and li 354: The data indicating an effect of HLA-dissimilarity on sperm velocity are relatively weak, being evident at only one time point. I am not convinced that this is sufficient to draw any firm conclusion on effects of HLA dissimilarity on motility. Why would it be advantageous to slow down the sperm of males that are 'overly' MHC dissimilar? Is there any evidence of any effect of "too much" dissimilarity in human partner selection? I would reconsider including this inference.

Review form: Reviewer 2

Recommendation

Major revision is needed (please make suggestions in comments)

Scientific importance: Is the manuscript an original and important contribution to its field?

Good

General interest: Is the paper of sufficient general interest?

Good

Quality of the paper: Is the overall quality of the paper suitable?

Acceptable

Is the length of the paper justified?

Yes

Should the paper be seen by a specialist statistical reviewer?

No

Do you have any concerns about statistical analyses in this paper? If so, please specify them explicitly in your report.

Yes

It is a condition of publication that authors make their supporting data, code and materials available - either as supplementary material or hosted in an external repository. Please rate, if applicable, the supporting data on the following criteria.

Is it accessible?

No

Is it clear?

N/A

Is it adequate?

N/A

Do you have any ethical concerns with this paper?

No

Comments to the Author

In the manuscript 'The post-copulatory genetic matchmaking: HLA-dependent effects of cervical mucus on human sperm function' the authors report the results from an experiment designed to test whether HLA-similarity between a female's cervical mucus and a male's sperm influences sperm motility, hyperactivation and viability in humans. The idea is intriguing, and the experiment appears to have been designed very well. A very recently published paper also concludes that male-female interactions are important for sperm behaviour in humans (<https://royalsocietypublishing.org/doi/10.1098/rspb.2020.0805>).

However, my main issue is that I don't think that the analyses in their current state appropriately reflect the experimental design, and the questions the authors wish to address. This concern is further elaborated in my specific comments below. For me, addressing whether this is an issue was further hindered by the privacy requirements of the data. I don't understand why the data could not be made available anonymously (like in the above-mentioned paper).

A more minor point is that to the non-expert, it is not clear what the difference is between Grantham distance and allele sharing. Are the two not correlated to a fairly large degree? Including both increases the number of tests performed, and hence the potential for finding a significant effect, so I feel this needs to be justified better.

Two more rather minor points: In my view, the MS is missing acknowledgement and discussion (apart from ascertaining that males were normozoospermic) of the fact that the participants were of course a biased subset of the population. Likewise, I'd like to see some (brief) discussion of the effect of CM dilution. By diluting, are you not significantly altering the non-newtonian properties of the CM?

Below some specific comments and recommended edits for the MS:

Title: Remove the "The" or add "hypothesis"

Keywords: Add "MHC" to keywords, remove "sperm" and "HLA" as they are already in the title

Line 14: Saying this hypothesis is "unexplored" seems a bit overstated. Your own references (6, 19, 79) and other papers (e.g. <https://doi.org/10.1111/mec.14490> & <https://doi.org/10.1111/mec.14153>) have explored this idea. Although technically your

statement is not wrong as it focusses on humans, the wording is a bit too strong IMO.

Line 145 or elsewhere: Please give an indication roughly how many sperm were measured to obtain average VCL, % hyperactivated and % alive?

Line 153: Given that you have a very nice design with a technical replicate and a biological replicate (sub-samples), it would be interesting to see how repeatable the measurements (technical repeatability) and sub-samples (biological repeatability) are.

Line 182: Analysing percent hyperactivated and viability using LMM is not ideal. I think you want a binomial GLMM here. But beware of overdispersion!

Line 183: What's the rationale for using sub-sample as a fixed factor? I don't see how one would expect the two sub-samples to consistently differ in the same way. In fact, I think when using the appropriate random effects structure, this factor is not needed. If of interest, why not use order measured (you say you reversed between the two sub-samples) to look for order effects?

Line 185: What is needed in addition is a random effect for male x female x sub-sample, and ideally random slopes for individual sub-samples so as to not pseudo-replicate time effects. After all, you measure the same biological replicate (sub-sample) at three timepoints, which makes them non-independent.

And what about repeated measures? You need another random factor to account for measuring every male-female-timepoint combination twice. Sub-samples are true replicates (within a given time-point), but these repeated measures are not.

Line 198: Removing this random slope when non-significant is not ideal. Generally, the model selection approach, although unfortunately still widely used, is subject to a phenomenon known as Winner's curse (<https://link.springer.com/article/10.1007/s00265-010-1038-5>).

Line 223: You show no effect sizes for fixed effects (and whether they are positive or negative) in your tables. Also, as mentioned elsewhere, to me some random effects pertaining to avoiding pseudoreplication are missing. Combined with the data not being available to check this myself, I find it quite hard to evaluate whether these models are appropriate.

Line 237: I wonder if this is the best way to show these interaction effects. This is merely a suggestion: Have you tried e.g. a 8 x 9 grid system (males as rows, females as columns or vice versa), with the value (here % hyperactivated as a colour (similar to gene expression heatmaps)). It would certainly make for a more compact figure...

Line 239: How is the SE calculated here? 2 sub-samples x 2 measurements?

Line 244: ditto line 237 and 239

Line 262: The details of these models are not presented. Given that to my understanding certain random effects are missing from the models, and that in the methods you mention the removal of non-significant effects (both fixed AND random), it is very difficult to judge the validity of this result. At least from Figure 3 it looks like there are more than two points from each male-female combination. Is this from replicates and repeated measures pooled? Were these repeated measures appropriately accounted for with random effects?

Line 319: An explicit test for this would be a quadratic effect of HLA-dissimilarity. Could you test for such an effect?

~Line 337: I was curious to see how the authors would relate their findings to Wedekind et al.'s study on mice, but although the paper is in the reference list (79), I couldn't find it cited in the

MS!?

Line 350f: While I agree in principle that genetic compatibility could be important, I don't think the data at hand provide strong evidence in support. The overall slope in Figure 3 shows an increase of only a few percent viability increase over the full range of HLA dissimilarity, which seems rather too subtle to cause major reproductive incompatibility by itself.

Decision letter (RSPB-2020-1060.R0)

18-Jun-2020

Dear Ms Jokiniemi:

I am writing to inform you that your manuscript RSPB-2020-1060 entitled "THE POST-COPULATORY GENETIC MATCHMAKING: HLA-DEPENDENT EFFECTS OF CERVICAL MUCUS ON HUMAN SPERM FUNCTION" has, in its current form, been rejected for publication in Proceedings B.

This action has been taken on the advice of referees, who have recommended that substantial revisions are necessary. With this in mind we would be happy to consider a resubmission, provided the comments of the referees are fully addressed. However please note that this is not a provisional acceptance.

Sincerely,
Dr Locke Rowe
<mailto:proceedingsb@royalsociety.org>

Associate Editor
Board Member: 1
Comments to Author:

This manuscript demonstrates that sperm performance (velocity, activation, and viability) is significantly influenced by the interaction between male and female identity in humans.

Interestingly, viability was correlated with HLA dissimilarity between males and females, but not overall genetic dissimilarity. This is consistent with cryptic female choice for genetically compatible mates. This manuscript is of potentially high significance since it offers some of the first empirical proof for a plausible mechanism behind cryptic female choice in humans.

Both reviewers provided a rather positive evaluation of the manuscript, and reviewer 1 mainly felt that certain statements need to be toned down or removed. However reviewer 2 had some concerns about the statistical analysis which I think need to be addressed. Reviewer 2 also specifically recommends against model simplification. I do not share this concern wholeheartedly, since the full model may have reduced power to detect significant effects due to over-parameterization. However I do feel that model simplification based on p-values alone is potentially problematic, and would prefer e.g. AIC values or some other method that does not involve hypothesis testing per se. Regardless, some re-analysis of the data will be necessary to address all of reviewer 2's concerns, which is why I am recommending rejection with possibility to resubmit.

Reviewer(s)' Comments to Author:

Referee: 1

Comments to the Author(s)

This manuscript investigates the physiological role and significance of cervical mucus-mediated effects on survival and function of sperm. The possibility of between individual female variance, between male variance, and interactions between male-female interactions were investigated in a full factorial design using clinical materials from 9 females and 8 males, and evaluated by linear mixed-model analysis. Compelling evidence of individual female-specific effects of cervical mucus on sperm motility, hyperactivation and viability were shown, in a male-partner specific manner. Additionally, sequencing analysis showed a clear relationship with extent of HLA disparity between male and female partners, indicating a female HLA-mediated mechanism.

This is an exciting and novel study that identifies a new mechanism by which post-copulatory cryptic female choice might operate in humans. The finding is important as, if a comparable process occurred in vivo, it reasonably would contribute to selective mechanisms impacting the likelihood of conception in a partner-specific manner. The study is well-designed, the analysis appears appropriate, and in general, the conclusions drawn are justified by the data provided. The paper is clearly and succinctly written, and relevant previous works are appropriately cited.

There are some issues listed below that if addressed, would improve the quality of the manuscript.

1. Abstract, li 24: "these results...indicate that the definitive mating bias towards HLA-dissimilar partners may occur after copulation". This sentence should be revised to recognise that other potential mechanisms for HLA-mediated mating bias may occur (these include pre-coital mechanisms such as pheromone-based partner selection, and perhaps other post-coital mechanisms operating at the level of embryo implantation). Consider "...indicate that one mechanism of bias towards HLA-dissimilar partners may occur after copulation".
2. Abstract, li 26 and discussion, li 350-351: "present findings raise the intriguing possibility that infertility may not represent exclusively a pathological condition". I recommend removing this statement from both the abstract and the discussion. Although I agree that here are likely biological mechanisms for increasing the prospects of successful mating depending on partner compatibility, and the current data indicate a new mechanism that could contribute to this, the study does not provide any evidence that might be construed to mean the selective mechanism is of sufficient force to achieve complete infertility. In fact, the data do not even show that selection occurs in vivo, so it is a very long bow to draw to impute a causal role in infertility. Moreover, the issue of infertility as pathology is a fraught and complex debate, and I do not agree that the current data add to the evidence either way.

3. Introduction, li 51: The authors state that HLA molecules are present on sperm and give a reference (Sereshki et al. 2019). The cited paper shows FACS data indicating less than 20% of sperm express HLAI or HLAI, and FACS plots are not particularly convincing. There has been considerable debate on this topic for decades, with several studies supporting both points of view. It should be clear in the authors comments that there is data both ways and the issue remains contentious.
4. Introduction, li 75: "cervical mucus contains soluble HLA and their antibodies". It is my understanding that anti-HLA antibodies are only rarely detected in female tract secretions, and generally in the context of infertility. The authors should qualify this statement accordingly.
5. Introduction, li 96: "We hypothesised that sperm selection.... occurs in the cervix". This aspect of the hypothesis was not tested, as no in vivo experiments were conducted. Could the hypothesis be rephased to read "is influenced by cervical mucus", as this is what was tested?
6. Methods, li 119: What was the minimum number of days of abstinence? This should be added.
7. Methods li 127: Did the authors consider repeating the experiment with the same biological materials on more than one occasion, to evaluate between experiment variance? Did they consider testing the consistency of the male-female relationships over time, by collecting semen and/or mucus from the same individuals on more than one occasion? This would improve confidence in the reliability of the findings.
8. Discussion, li 339: The authors comment on possible mechanisms by which sperm-specific effects might be mediated, and imply that sperm-borne HLA and/or anti-HLA antibodies could contribute. Is this possible if only a very small proportion of sperm express HLAI and/or HLAI (see above), or if such antibodies are only present in a subset of women? Did they consider measuring these in their cervical mucus samples and relating the presence of antibodies to the male-specific effects? An additional comment on the limitations of this speculation would be warranted.
9. Discussion, li 321 and li 354: The data indicating an effect of HLA-dissimilarity on sperm velocity are relatively weak, being evident at only one time point. I am not convinced that this is sufficient to draw any firm conclusion on effects of HLA dissimilarity on motility. Why would it be advantageous to slow down the sperm of males that are 'overly' MHC dissimilar? Is there any evidence of any effect of "too much" dissimilarity in human partner selection? I would reconsider including this inference.

Referee: 2

Comments to the Author(s)

In the manuscript 'The post-copulatory genetic matchmaking: HLA-dependent effects of cervical mucus on human sperm function' the authors report the results from an experiment designed to test whether HLA-similarity between a female's cervical mucus and a male's sperm influences sperm motility, hyperactivation and viability in humans. The idea is intriguing, and the experiment appears to have been designed very well. A very recently published paper also concludes that male-female interactions are important for sperm behaviour in humans (<https://royalsocietypublishing.org/doi/10.1098/rspb.2020.0805>).

However, my main issue is that I don't think that the analyses in their current state appropriately reflect the experimental design, and the questions the authors wish to address. This concern is further elaborated in my specific comments below. For me, addressing whether this is an issue was further hindered by the privacy requirements of the data. I don't understand why the data could not be made available anonymously (like in the above-mentioned paper).

A more minor point is that to the non-expert, it is not clear what the difference is between Grantham distance and allele sharing. Are the two not correlated to a fairly large degree? Including both increases the number of tests performed, and hence the potential for finding a significant effect, so I feel this needs to be justified better.

Two more rather minor points: In my view, the MS is missing acknowledgement and discussion (apart from ascertaining that males were normozoospermic) of the fact that the participants were

of course a biased subset of the population. Likewise, I'd like to see some (brief) discussion of the effect of CM dilution. By diluting, are you not significantly altering the non-newtonian properties of the CM?

Below some specific comments and recommended edits for the MS:

Title: Remove the "The" or add "hypothesis"

Keywords: Add "MHC" to keywords, remove "sperm" and "HLA" as they are already in the title

Line 14: Saying this hypothesis is "unexplored" seems a bit overstated. Your own references (6, 19, 79) and other papers (e.g. <https://doi.org/10.1111/mec.14490> & <https://doi.org/10.1111/mec.14153>) have explored this idea. Although technically your statement is not wrong as it focusses on humans, the wording is a bit too strong IMO.

Line 145 or elsewhere: Please give an indication roughly how many sperm were measured to obtain average VCL, % hyperactivated and % alive?

Line 153: Given that you have a very nice design with a technical replicate and a biological replicate (sub-samples), it would be interesting to see how repeatable the measurements (technical repeatability) and sub-samples (biological repeatability) are.

Line 182: Analysing percent hyperactivated and viability using LMM is not ideal. I think you want a binomial GLMM here. But beware of overdispersion!

Line 183: What's the rationale for using sub-sample as a fixed factor? I don't see how one would expect the two sub-samples to consistently differ in the same way. In fact, I think when using the appropriate random effects structure, this factor is not needed. If of interest, why not use order measured (you say you reversed between the two sub-samples) to look for order effects?

Line 185: What is needed in addition is a random effect for male x female x sub-sample, and ideally random slopes for individual sub-samples so as to not pseudo-replicate time effects. After all, you measure the same biological replicate (sub-sample) at three timepoints, which makes them non-independent.

And what about repeated measures? You need another random factor to account for measuring every male-female-timepoint combination twice. Sub-samples are true replicates (within a given time-point), but these repeated measures are not.

Line 198: Removing this random slope when non-significant is not ideal. Generally, the model selection approach, although unfortunately still widely used, is subject to a phenomenon known as Winner's curse (<https://link.springer.com/article/10.1007/s00265-010-1038-5>).

Line 223: You show no effect sizes for fixed effects (and whether they are positive or negative) in your tables. Also, as mentioned elsewhere, to me some random effects pertaining to avoiding pseudoreplication are missing. Combined with the data not being available to check this myself, I find it quite hard to evaluate whether these models are appropriate.

Line 237: I wonder if this is the best way to show these interaction effects. This is merely a suggestion: Have you tried e.g. a 8 x 9 grid system (males as rows, females as columns or vice versa), with the value (here % hyperactivated as a colour (similar to gene expression heatmaps). It would certainly make for a more compact figure...

Line 239: How is the SE calculated here? 2 sub-samples x 2 measurements?

Line 244: ditto line 237 and 239

Line 262: The details of these models are not presented. Given that to my understanding certain random effects are missing from the models, and that in the methods you mention the removal of non-significant effects (both fixed AND random), it is very difficult to judge the validity of this result. At least from Figure 3 it looks like there are more than two points from each male-female combination. Is this from replicates and repeated measures pooled? Were these repeated measures appropriately accounted for with random effects?

Line 319: An explicit test for this would be a quadratic effect of HLA-dissimilarity. Could you test for such an effect?

~Line 337: I was curious to see how the authors would relate their findings to Wedekind et al.'s study on mice, but although the paper is in the reference list (79), I couldn't find it cited in the MS!?

Line 350f: While I agree in principle that genetic compatibility could be important, I don't think the data at hand provide strong evidence in support. The overall slope in Figure 3 shows an increase of only a few percent viability increase over the full range of HLA dissimilarity, which seems rather too subtle to cause major reproductive incompatibility by itself.

Author's Response to Decision Letter for (RSPB-2020-1060.R0)

See Appendix A.

RSPB-2020-1682.R0

Review form: Reviewer 2

Recommendation

Accept with minor revision (please list in comments)

Scientific importance: Is the manuscript an original and important contribution to its field?

Good

General interest: Is the paper of sufficient general interest?

Good

Quality of the paper: Is the overall quality of the paper suitable?

Good

Is the length of the paper justified?

Yes

Should the paper be seen by a specialist statistical reviewer?

No

Do you have any concerns about statistical analyses in this paper? If so, please specify them explicitly in your report.

No

It is a condition of publication that authors make their supporting data, code and materials available - either as supplementary material or hosted in an external repository. Please rate, if applicable, the supporting data on the following criteria.

Is it accessible?

No

Is it clear?

N/A

Is it adequate?

N/A

Do you have any ethical concerns with this paper?

No

Comments to the Author

By and large, all my previous comments have been addressed satisfactorily by the authors. Great to see that the authors got some additional advice on the random effects structure, which now looks adequate.

There are very few remaining issues that I think would benefit from clarification:

Figure 3: "The figure shows modelled datapoints obtained from the LMM." It is not clear to me what the authors mean. Perhaps they could explain this in a bit more detail?

It would be great if the authors could include (as online supplementary material) overviews of the AIC-based model comparisons. I understand that the authors selected the models with the best AIC, but it could be of interest to some readers to see how strongly models differed (i.e. whether there were competing models with similar AIC values).

Decision letter (RSPB-2020-1682.R0)

20-Jul-2020

Dear Ms Jokiniemi

I am pleased to inform you that your manuscript RSPB-2020-1682 entitled "POST-COPULATORY GENETIC MATCHMAKING: HLA-DEPENDENT EFFECTS OF CERVICAL MUCUS ON HUMAN SPERM FUNCTION" has been accepted for publication in Proceedings B.

The referee(s) have recommended publication, but also suggest some minor revisions to your manuscript. Therefore, I invite you to respond to the referee(s)' comments and revise your manuscript. Because the schedule for publication is very tight, it is a condition of publication that you submit the revised version of your manuscript within 7 days. If you do not think you will be able to meet this date please let us know.

To revise your manuscript, log into <https://mc.manuscriptcentral.com/prsb> and enter your Author Centre, where you will find your manuscript title listed under "Manuscripts with Decisions." Under "Actions," click on "Create a Revision." Your manuscript number has been appended to denote a revision. You will be unable to make your revisions on the originally

submitted version of the manuscript. Instead, revise your manuscript and upload a new version through your Author Centre.

Sincerely,
Dr Locke Rowe
mailto: proceedingsb@royalsociety.org

Associate Editor
Comments to Author:

The authors have done a satisfactory job of responding to all of the comments raised in the previous round of review. Reviewer 1 (previously reviewer 2) just has a few minor points that can be addressed before acceptance.

Reviewer(s)' Comments to Author:

Referee: 2

Comments to the Author(s).

By and large, all my previous comments have been addressed satisfactorily by the authors. Great to see that the authors got some additional advice on the random effects structure, which now looks adequate.

There are very few remaining issues that I think would benefit from clarification:

Figure 3: "The figure shows modelled datapoints obtained from the LMM." It is not clear to me what the authors mean. Perhaps they could explain this in a bit more detail?

It would be great if the authors could include (as online supplementary material) overviews of the AIC-based model comparisons. I understand that the authors selected the models with the best AIC, but it could be of interest to some readers to see how strongly models differed (i.e. whether there were competing models with similar AIC values).

Decision letter (RSPB-2020-1682.R1)

27-Jul-2020

Dear Ms Jokiniemi

I am pleased to inform you that your manuscript entitled "POST-COPULATORY GENETIC MATCHMAKING: HLA-DEPENDENT EFFECTS OF CERVICAL MUCUS ON HUMAN SPERM FUNCTION" has been accepted for publication in Proceedings B.

You can expect to receive a proof of your article from our Production office in due course, please check your spam filter if you do not receive it. PLEASE NOTE: you will be given the exact page

length of your paper which may be different from the estimation from Editorial and you may be asked to reduce your paper if it goes over the 10 page limit.

Open Access

Paper charges

Sincerely,

Proceedings B

Appendix A

Dear Editors and Reviewers,

We would like to thank you all for your constructive criticism towards our manuscript. We think that this feedback allowed us to significantly improve our paper. Below you can find our detailed reply to each individual comment (in bold font). Required modifications are also highlighted in red font in our revised manuscript.

Associate Editor

Board Member: 1

Comments to Author:

This manuscript demonstrates that sperm performance (velocity, activation, and viability) is significantly influenced by the interaction between male and female identity in humans. Interestingly, viability was correlated with HLA dissimilarity between males and females, but not overall genetic dissimilarity. This is consistent with cryptic female choice for genetically compatible mates. This manuscript is of potentially high significance since it offers some of the first empirical proof for a plausible mechanism behind cryptic female choice in humans.

Both reviewers provided a rather positive evaluation of the manuscript, and reviewer 1 mainly felt that certain statements need to be toned down or removed. However reviewer 2 had some concerns about the statistical analysis which I think need to be addressed. Reviewer 2 also specifically recommends against model simplification. I do not share this concern whole-heartedly, since the full model may have reduced power to detect significant effects due to over-parameterization. However I do feel that model simplification based on p-values alone is potentially problematic, and would prefer e.g. AIC values or some other method that does not involve hypothesis testing per se. Regardless, some re-analysis of the data will be necessary to address all of reviewer 2's concerns, which is why I am recommending rejection with possibility to resubmit.

Reply: We have now significantly toned down our claims and re-structured our statistical models, as requested by the reviewers. Although we feel that reviewer # 2 statistical criticism is largely valid, we also agree with the associate editor that it is not possible to follow “keep it maximal” principle (see Barr et al. 2013, Journal of Memory and Language 68: 255–278) in all the situations, for example when the full models do not converge. In fact, the whole principle whether to keep maximal or simplified final models is hotly debated and for example Matuschek et al. 2017 (Journal of Memory and Language 94:305–315) recommend simplified models, since such models have higher power, but still reasonable low type I error rate. When fitted our revised models (see detailed explanation below), we confronted some relatively rare convergence problems and in these cases were obliged to simplify the models based on AIC. Similarly (and again based on AIC), we removed clearly redundant interaction effects from some of our models that did not improve model fit (see revised manuscript), and thus reduced the risk of over-parameterization. In other words, model selection is now based solely on AIC values, not P-values as in our original version of the manuscript. Importantly, despite we re-structured our models, all our main results remained the same, meaning that also all our original conclusions are still valid. We have, however, revised our statistical results to take into account reviewer’s comments.

Reviewer(s)' Comments to Author:

Referee: 1

Comments to the Author(s)

This manuscript investigates the physiological role and significance of cervical mucus-mediated effects on survival and function of sperm. The possibility of between individual female variance, between male variance, and interactions between male-female interactions were investigated in a full factorial design using clinical materials from 9 females and 8 males, and evaluated by linear mixed-model analysis. Compelling evidence of individual female-specific effects of cervical mucus on sperm motility, hyperactivation and viability were shown, in a male-partner specific manner. Additionally, sequencing analysis showed a clear relationship with extent of HLA disparity between male and female partners, indicating a female HLA-mediated mechanism.

This is an exciting and novel study that identifies a new mechanism by which post-copulatory cryptic female choice might operate in humans. The finding is important as, if a comparable process occurred in vivo, it reasonably would contribute to selective mechanisms impacting the likelihood of conception in a partner-specific manner. The study is well-designed, the analysis appears appropriate, and in general, the conclusions drawn are justified by the data provided. The paper is clearly and succinctly written, and relevant previous works are appropriately cited.

There are some issues listed below that if addressed, would improve the quality of the manuscript.

1. Abstract, li 24: “these results...indicate that the definitive mating bias towards HLA-dissimilar partners may occur after copulation”. This sentence should be revised to recognise that other potential mechanisms for HLA-mediated mating bias may occur (these include pre-coital mechanisms such as pheromone-based partner selection, and perhaps other post-coital mechanisms operating at the level of embryo implantation). Consider “...indicate that one mechanism of bias towards HLA-dissimilar partners may occur after copulation”.

Reply: We agree with the reviewer that other potential forms of HLA-mediated mating bias may occur as well. Thus, we have modified this sentence and it now states: “...processes occurring after copulation may contribute to the mating bias towards HLA dissimilar partners (see lines 24-25).

2. Abstract, li 26 and discussion, li 350-351: “present findings raise the intriguing possibility that infertility may not represent exclusively a pathological condition”. I recommend removing this statement from both the abstract and the discussion. Although I agree that here are likely biological mechanisms for increasing the prospects of successful mating depending on partner compatibility, and the current data indicate a new mechanism that could contribute to this, the study does not provide any evidence that might be construed to mean the selective mechanism is of sufficient force to achieve complete infertility. In fact, the data do not even show that selection occurs in vivo, so it is a very long bow to draw to impute a causal role in infertility. Moreover, the issue of infertility as pathology

is a fraught and complex debate, and I do not agree that the current data add to the evidence either way.

Reply: We agree with the reviewer that our findings do not provide definitive evidence that partner incompatibility could lead to infertility. We also agree that genetic incompatibility at the gamete-level may not frequently lead to complete infertility, but more likely reduce the probability of conception. This has now been highlighted more clearly in the revised manuscript (lines 397-399). We also removed suggested statements from our revised manuscript. Reviewer seems to share our view that mating success can be dependent on partner (in)compatibility and that our results could offer novel mechanistic explanation for this process. Thus, we now highlight that it is possible that our results provide novel insights into the deeper understanding of infertility. Given that there is considerable amount of evidence from various non-human animals (also in vivo) that sperm physiological response to various female-derived chemical factors predict later fertilization success (or failure), we feel that highlighting this possibility in humans is a valid argument. Supporting this view, it has been shown that in humans, sperm performance in cervical mucus predicts fertilization success (see lines 346-347). Based on these facts and to encourage future studies to further clarify this possibility, we would not completely reject the idea that gamete-level incompatibility may play some role behind infertility problems. In any case, to take into account reviewer's concerns, we have significantly toned our earlier claims in the abstract and discussion (lines 25-27 and 397-399).

3. Introduction, li 51: The authors state that HLA molecules are present on sperm and give a reference (Sereshki et al. 2019). The cited paper shows FACS data indicating less than 20% of sperm express HLA I or HLA II, and FACS plots are not particularly convincing. There has been considerable debate on this topic for decades, with several studies supporting both points of view. It should be clear in the authors comments that there is data both ways and the issue remains contentious.

Reply: We agree that the HLA expression in sperm is still under debate and have now included references from both sides of the debate (see lines 51-52).

4. Introduction, li 75: "cervical mucus contains soluble HLA and their antibodies". It is my understanding that anti-HLA antibodies are only rarely detected in female tract secretions, and generally in the context of infertility. The authors should qualify this statement accordingly.

Reply: Yes, anti-HLA antibodies are frequently linked with infertility and are thus likely present only a subset of females. As it is unlikely that these antibodies can explain our findings, we removed this mention from our revised manuscript.

5. Introduction, li 96: "We hypothesised that sperm selection... occurs in the cervix". This aspect of the hypothesis was not tested, as no in vivo experiments were conducted. Could the hypothesis be rephased to read "is influenced by cervical mucus", as this is what was tested?

Reply: We have now rephrased the hypothesis as suggested (lines 97-98).

6. Methods, li 119: What was the minimum number of days of abstinence? This should be added.

Reply: The minimum abstinence was one day. We asked all male donors to abstain at least two days, but in one male abstinence period was only one day. Since the sperm concentration in this donor was well sufficient for our analyses, we decided to include this sample. We have now included this information in the revised manuscript (line 119).

7. Methods li 127: Did the authors consider repeating the experiment with the same biological materials on more than one occasion, to evaluate between experiment variance? Did they consider testing the consistency of the male-female relationships over time, by collecting semen and/or mucus from the same individuals on more than one occasion? This would improve confidence in the reliability of the findings.

Reply: Surely investigating whether our results are consistent over time would be important. However, due to the ethical and many practical issues, this would be challenging to realize in humans. In Finland, the collection of cervical mucus is a clinical procedure that (in our study design) can only be conducted during the (induced) ovulation and is performed only by medical experts. Furthermore, recruiting females for cervical mucus donation was proven to be rather challenging even if we asked females to donate only one sample. Thus, collecting two or more temporal replicate samples would have required plenty of extra work and special arrangements. Since the primary aim of our study was to demonstrate whether the whole phenomenon (cryptic female choice) could exist in humans, for the above-mentioned reasons, we did not consider testing consistency of the male-female relationships over time at this stage. Instead, we paid special attention to within experiment variation by measuring sperm motility in three different time points and by analyzing sperm traits in four replicate measurements in each male-female combination (and timepoint). Supporting the validity of our results, our sperm measurements were highly repeatable both between sub-samples and between replicate measurements within sub-samples (now mentioned in lines 238-240).

8. Discussion, li 339: The authors comment on possible mechanisms by which sperm-specific effects might be mediated, and imply that sperm-borne HLA and/or anti-HLA antibodies could contribute. Is this possible if only a very small proportion of sperm express HLA I and/or HLA II (see above), or if such antibodies are only present in a subset of women? Did they consider measuring these in their cervical mucus samples and relating the presence of antibodies to the male-specific effects? An additional comment on the limitations of this speculation would be warranted.

Reply: As mentioned above (reviewer's comment 4), we agree that anti-HLA antibodies are likely present only a subset of females and thus have removed this speculation from our revised discussion as well. However, we still tend to think that even if HLA class I and II proteins would be present only in a small proportion of sperm, these molecules may potentially play an important role behind our findings. This is because in humans only about 10-20% of sperm are functionally capable ('capacitated') to fertilize the eggs at given timepoint, which is roughly the same than the proportion of HLA positive cells found by Sereshki et al. (2019). Although we cannot show any evidence for our hypothesis at this stage, we are wondering whether only capacitated or 'fertilization-ready' sperm cells express HLAs. In theory this is possible as for example capacitation is associated with the release of numerous sperm surface proteins (see e.g. Hernández-Silva et al. 2020, *Andrology* 8: 171-180), which potentially can conceal underlying HLAs in incapacitated sperm. When this physiological feature of the human sperm and associated changes in sperm plasma membrane are considered, we do not necessarily see strong

reasons to assume that all the human sperm should express HLAs at the same time. In our view, it is also possible that failure to study sperm in functionally relevant physiological state (incapacitated sperm without the presence of female-derived chemical factors) may at least partly explain why so many studies have failed to demonstrate sperm HLA expression. Even if we find this possible, we would not discuss these possibilities in the manuscript as we lack experimental support for these claims. However, based on these possibilities we assume that low prevalence of HLA positive sperm cells does not necessarily mean that sperm surface HLAs cannot play an important role in sperm selection and gamete-level compatibility.

Our original plan was to collect enough cervical mucus for later immunological analyses. Unfortunately, this was not possible for most of the females, as the volume of the cervical mucus varies considerably, and the remaining cervical mucus samples have already been used in another study. However, we agree with the reviewer, that it would be important to study the presence of anti-HLA antibodies in cervical mucus. Thus, this is clearly an important task for future studies. As we mention above, we have removed all the speculation concerning anti-HLA antibodies from our revised manuscript.

9. Discussion, li 321 and li 354: The data indicating an effect of HLA-dissimilarity on sperm velocity are relatively weak, being evident at only one time point. I am not convinced that this is sufficient to draw any firm conclusion on effects of HLA dissimilarity on motility. Why would it be advantageous to slow down the sperm of males that are ‘overly’ MHC dissimilar? Is there any evidence of any effect of “too much” dissimilarity in human partner selection? I would reconsider including this inference.

Reply: We agree with the reviewer that we cannot make firm conclusions on the effect of HLA dissimilarity on sperm motility and have already highlighted this in the manuscript (see lines 370-372). We have also further clarified our rationale behind selection against ‘overly’ similar and dissimilar males in general and in humans, as suggested by the reviewer (lines 362-370).

Choosing a partner with whom the female shares an intermediate number of MHC alleles is thought to be beneficial by allowing a balance between inbreeding and outbreeding costs. Moreover, according to the optimal MHC-heterozygosity hypothesis, the best immunocompetence would not be achieved with complete heterozygosity, but with a lower level of heterozygosity (Penn and Potts 1999, see our manuscript for full reference). This is because highly heterozygous individuals at MHC loci present more antigens to the immune system, but probably have smaller T-cell repertoires following thymic selection. Indeed, it has been shown in some fish species that females prefer males that have intermediate number of shared MHC alleles (e.g. Lenz et al. 2018; Radwan et al. 2020). There is also some evidence for such a preference for intermediate HLA allele number in humans (although other studies reported preferences for HLA dissimilar partners: e.g. Wedekind et al. 1995). In fact, Jacob et al. (2002) showed that among hutterites, women prefer the odor of men with whom they share few HLA alleles, compared to men with whom they do not share any allele or with whom they share many. We included the reference to Jacob et al. in the revised manuscript and clarified how an intermediate HLA dissimilarity could be beneficial (lines 362-370).

Referee: 2

Comments to the Author(s)

In the manuscript 'The post-copulatory genetic matchmaking: HLA-dependent effects of cervical mucus on human sperm function' the authors report the results from an experiment designed to test whether HLA-similarity between a female's cervical mucus and a male's sperm influences sperm motility, hyperactivation and viability in humans. The idea is intriguing, and the experiment appears to have been designed very well. A very recently published paper also concludes that male-female interactions are important for sperm behaviour in humans

(<https://royalsocietypublishing.org/doi/10.1098/rspb.2020.0805>).

Reply: Indeed, this is a very important and interesting paper and we have now cited it in our revised manuscript (line 45).

However, my main issue is that I don't think that the analyses in their current state appropriately reflect the experimental design, and the questions the authors wish to address. This concern is further elaborated in my specific comments below. For me, addressing whether this is an issue was further hindered by the privacy requirements of the data. I don't understand why the data could not be made available anonymously (like in the above-mentioned paper).

Reply: We thank reviewer for these important comments. Our detailed reply can be found below. Unfortunately, our ethical permit and the terms of the consent do not allow us to publicly share our individual-level data. We are not able to evaluate the reasons why the data by Fitzpatrick et al. has been made publicly available, but one potential reason for this may be that their data does not include any genetic information. According to our local ethics committee (Ethics Committee, Hospital District of Northern Savo, Finland), genetic data may allow identification of the study subjects even without any additional personal data (and even if we have pseudonymized our data). Although we don't necessarily fully agree, we are obliged to follow their instructions.

A more minor point is that to the non-expert, it is not clear what the difference is between Grantham distance and allele sharing. Are the two not correlated to a fairly large degree? Including both increases the number of tests performed, and hence the potential for finding a significant effect, so I feel this needs to be justified better.

Reply: Grantham distance and allele sharing are indeed somewhat correlated, but still measure different aspects of HLA dissimilarity. Grantham distance considers the molecular volume of amino acids composing different HLA alleles, which is important to determine which peptides will bind to the peptide-binding groove of the HLA molecule. Consequently, in comparison to allele sharing, Grantham distance is expected to be more relevant to describe the functional properties of the HLA alleles and thus to represent a more informative (functional) measure of HLA dissimilarity. This was our primary reason to include Grantham distance in our analyses. On the other hand, HLA allele sharing was also included to make our results directly comparable to previous studies, which often have reported only this more traditional measure of HLA similarity. We have now included a more detailed explanation on Grantham distance to further clarify the difference between these parameters (lines 176-178).

Two more rather minor points: In my view, the MS is missing acknowledgement and discussion (apart from ascertaining that males were normozoospermic) of the fact that the participants were of course a biased subset of the population. Likewise, I'd like to see some (brief) discussion of the effect of CM dilution. By diluting, are you not significantly altering the non-newtonian properties of the CM?

Reply: This is a relevant point and it is true that all the study subjects were recruited from the fertility clinic for compulsory practical reasons. On the other hand, what comes to males, in the recruited 'sub-population' the mean sperm concentration, motility and proportion of morphologically normal spermatozoa was likely slightly higher than on average 'male population'. Anyway, naturally for this same reason (and because we recruited also females from the same fertility clinic) we agree that recruited subjects did not likely fully represent larger human population and have now discussed this accordingly in our revised manuscript (lines 335-339).

Cervical mucus was diluted for the following reasons. First, our primary aim was to study, whether the chemical composition of cervical mucus could mediate cryptic female choice (highlighted e.g. in lines 87-88, 109). For this purpose, we needed to rupture the mucin network and dilute samples, which allowed us to separate these chemically mediated effects from the physical (structural) effects of the cervical mucus (see lines 89-91). Second, due to the low volume of collected cervical mucus samples, dilution was inevitable to increase the sample volume. Without this dilution we would not have enough cervical mucus to perform all required sperm treatments between numerous male-female combinations. Third, without dilution, it is practically impossible to mix the cervical mucus with the sperm and perform sperm motility and viability analyses. In fact, undiluted cervical mucus has such a high viscosity that it cannot be pipeted at all. In any case, while the alteration of cervical mucus viscoelastic properties is likely to influence sperm motility (although we expect it to have limited consequences on sperm viability), we would expect it to influence sperm from all males in a similar fashion. In other words, we find it unlikely that dilution would have had any biasing effect on observed male-female compatibility effects. We have now added some additional discussion concerning cervical mucus dilution, as suggested by the reviewer (lines 372-374).

Below some specific comments and recommended edits for the MS:

Title: Remove the "The" or add "hypothesis"

Reply: We have removed "The" from our title.

Keywords: Add "MHC" to keywords, remove "sperm" and "HLA" as they are already in the title

Reply: We have modified the keywords according to the reviewer's suggestion.

Line 14: Saying this hypothesis is "unexplored" seems a bit overstated. Your own references (6, 19, 79) and other papers (e.g. <https://doi.org/10.1111/mec.14490> & <https://doi.org/10.1111/mec.14153>) have explored this idea. Although technically your statement is not wrong as it focusses on humans, the wording is a bit too strong IMO.

Reply: We have modified our wording to highlight that our statement refers only to humans (lines 13-14). We also cited these two references, highlighted by the reviewer.

Line 145 or elsewhere: Please give an indication roughly how many sperm were measured to obtain average VCL, % hyperactivated and % alive?

Reply: We have included this information in the manuscript (lines 145-146 and 153-154).

Line 153: Given that you have a very nice design with a technical replicate and a biological replicate (sub-samples), it would be interesting to see how repeatable the measurements (technical repeatability) and sub-samples (biological repeatability) are.

Reply: All our measurements were highly repeatable, both between sub-samples and between measurements (= replicates) within the sub-samples, for all sperm parameters. We have now included these new results in our revised manuscript (lines 190-192 and 238-240).

Line 182: Analysing percent hyperactivated and viability using LMM is not ideal. I think you want a binomial GLMM here. But beware of overdispersion!

Reply: We agree that binomial GLMMs are good approach to model proportion data. Thus, based on reviewer's suggestion we also re-analyzed our data using binomial GLMMs (with observation-level random effects to deal with overdispersion). All our main results remained the same and thus both LMMs and GLMMs lead to identical conclusions. For this reason and due to the following facts, we would like to argue analyzing our data with LMMs.

First, in comparison to linear mixed models, GLMMs are computationally more complex, harder to diagnose (e.g. for overdispersion), and also do not readily allow determination of required variance components (i.e. proportion of total variance explained by random effects). One of our primary interests in this study was to accurately estimate the proportion of total variance explained by male-female interaction effect (see lines 234, 324, Tables 2-4). The fact that GLMMs do not have any residual variance make this much more challenging.

Second, based on the comments by the reviewer and associate editor, we have added additional random effects into our models (see below). Adding these additional effects occasionally led to model convergence problems, and we were obliged to simplify such models. Following associate editor's suggestion, we did this by using AIC. However, computation of AIC-values is not trivial in GLMMs, which may make AIC-values unreliable. These new random effects also contained < 5 levels, when the available methods, to account for overdispersion perform relatively poorly (Harrison et al. 2015, PeerJ 3: e1114).

Third, although GLMMs usually converged without problems, in one critical occasion (i.e. when testing for the association between HLA dissimilarity and sperm viability) the binomial model failed to converge, but the corresponding LMM converged without problems.

Finally, since our model diagnostics for LMMs suggests that all our models are adequate, for these practical reasons, we decided to keep LMMs also in our revised manuscript. We hope that reviewer understand our point of view.

Line 183: What's the rationale for using sub-sample as a fixed factor? I don't see how one would

expect the two sub-samples to consistently differ in the same way. In fact, I think when using the appropriate random effects structure, this factor is not needed. If of interest, why not use order measured (you say you reversed between the two sub-samples) to look for order effects?

Reply: We agree with the reviewer that this parameter (sub-sample) should not be included as a fixed factor and thus we now model it as a random factor (see our reply below). However, we still think that it cannot be removed completely for the following reasons. First, the effect of testing order is strongly dependent on female ID, as female samples were always tested in the same order (although the two subsamples, A and B, were tested in the opposite order). For this reason, including this suggested new parameter would result in the situation where it absorbs all the variation explained by female ID (i.e. female variance would become 0). Secondly, for each male's sperm, we initiated cervical mucus treatment in three minutes intervals (0 min = CM1, 3 min = CM2, ... , 24 min = CM9) first in the sub-sample A and then identically (but in reversed order) in sub-sample B (30 min = CM9, 33 min = CM8, ... , 54 min = CM1). Although the average initiation time for two sub-samples is the same (27 min after the first treatment: CM1) for all nine women, in the sub-sample B the sperm was on average 30 min older than in subsample A. By taking sub-sample as a random factor in our models, we are controlling this time effect. This 3 min delay between the samples accounted for the time needed to perform sperm motility and sperm viability measurements and it was practically impossible to avoid it. Since this aspect clearly was not highlighted accurately enough in the previous version of our manuscript, we have now further clarified our experimental approach (see lines 157-164).

Line 185: What is needed in addition is a random effect for male x female x sub-sample, and ideally random slopes for individual sub-samples so as to not pseudo-replicate time effects. After all, you measure the same biological replicate (sub-sample) at three timepoints, which makes them non-independent. And what about repeated measures? You need another random factor to account for measuring every male-female-timepoint combination twice. Sub-samples are true replicates (within a given time-point), but these repeated measures are not.

Reply: Regarding this comment we consulted Lauri Mehtätalo, a professor of applied statistics at UEF. Based on this consultation we agree with the reviewer that we need to model random slope of time for individual sub-samples and additionally for male, female and male-female interaction. Repeated measures within sub-samples are now accounted for by including random effect of time also for the interaction between sub-sample and replicate measurement: (Timepoint|sub-sample:replicate). In the timepoint-specific models we also included two new random effects: (1|sub-sample) and (1| sub-sample:replicate). Thus, as mentioned above, sub-sample is now modelled as a random effect (not fixed effect anymore) in timepoint-specific models as well. Although adding these new random effects complicated the models, large majority of the models converged without problems. If no convergence problems were detected, we either kept the full model or simplified it (based on AIC, as suggested by the associate editor) by removing interactions that did not improve model fit. If model failed to converge, we again simplified the model based on AIC. Regardless of these modifications, all the results remained qualitatively the same (see revised results section and tables). Based on these new models and reviewer's comments we have now re-written our statistical methods (lines 190-221).

Line 198: Removing this random slope when non-significant is not ideal. Generally, the model selection approach, although unfortunately still widely used, is subject to a phenomenon known as Winner's curse (<https://link.springer.com/article/10.1007/s00265-010-1038-5>).

Reply: In our revised manuscript we simplify the model only based on AIC, not P-values anymore as we did earlier. We removed redundant interactions between timepoint and genetic parameters and between male and genetic parameter, based on AIC values. In other words, these interactions were removed only if they did not improve model fit. Additionally, as we explained above, we simplified our model if the full model did not converge and did not use any stepwise model selection approaches, which indeed are widely criticized. We thank reviewer for the link to this interesting article.

Line 223: You show no effect sizes for fixed effects (and whether they are positive or negative) in your tables. Also, as mentioned elsewhere, to me some random effects pertaining to avoiding pseudoreplication are missing. Combined with the data not being available to check this myself, I find it quite hard to evaluate whether these models are appropriate.

Reply: As explained above, we have now modified our statistical analyses to avoid pseudoreplication. We have now included modified Table 1, where we have replaced *F*-statistics with *t*-statistics, which gives both the intercept and the sign (+/-) of the slope for the fixed effects. This change does not affect our results.

Line 237: I wonder if this is the best way to show these interaction effects. This is merely a suggestion: Have you tried e.g. a 8 x 9 grid system (males as rows, females as columns or vice versa), with the value (here % hyperactivated as a colour (similar to gene expression heatmaps). It would certainly make for a more compact figure...

Reply: We tried to make heatmaps as suggested. Generally, we like the idea, but still find heatmaps a bit problematic as they don't allow including any estimate for the variance (just mean value, indicated by different hues). Thus, we would still keep our original figures, if possible, as we find them more informative in this sense.

Line 239: How is the SE calculated here? 2 sub-samples x 2 measurements?

Reply: Yes, this is correct. We now specify this in the figure caption (line 272).

Line 244: ditto line 237 and 239

Reply: Yes, this is correct. We now specify this in the figure caption (line 288).

Line 262: The details of these models are not presented. Given that to my understanding certain random effects are missing from the models, and that in the methods you mention the removal of non-significant effects (both fixed AND random), it is very difficult to judge the validity of this result. At least from Figure 3 it looks like there are more than two points from each male-female combination. Is this from replicates and repeated measures pooled? Were these repeated measures appropriately accounted for with random effects?

Reply: Sperm viability was measured only in one timepoint, so it was modelled by using identical model structure than in timepoint-specific models (see our reply above: "Line 185").

In other words, we have now added these ‘missing’ random effects. Figure 3 shows in total of four replicate measurements (i.e. two replicate measurements within both sub-samples) per male-female combination (now explained in the figure caption).

Line 319: An explicit test for this would be a quadratic effect of HLA-dissimilarity. Could you test for such an effect?

Reply: Here we hypothesize that cervical mucus could mediate sperm selection by acting both on sperm viability and motility, but in opposite directions: overly HLA similar males may be selected against via reduced viability, and overly dissimilar via reduced motility. Since this suggested effect is based on two distinct parameters, it cannot be analyzed as suggested.

~Line 337: I was curious to see how the authors would relate their findings to Wedekind et al.’s study on mice, but although the paper is in the reference list (79), I couldn’t find it cited in the MS!?

Reply: We thank reviewer for highlighting this, as this essential reference indeed was missing from the main text. Thus, we added citation to Wedekind et al. (1996) in the introduction (line 41), and also added a sentence in the discussion to relate our findings to previous work on MHC-based cryptic female choice, where we cite this paper another time (line 349-352).

Line 350f: While I agree in principle that genetic compatibility could be important, I don’t think the data at hand provide strong evidence in support. The overall slope in Figure 3 shows an increase of only a few percent viability increase over the full range of HLA dissimilarity, which seems rather too subtle to cause major reproductive incompatibility by itself.

Reply: We would like to point out that the Figure 3 actually do not show the full range of HLA dissimilarity as our data did not include male-female combinations that would have had 0 to 3 dissimilar alleles. Thus, in the whole dissimilarity range we could expect to see higher viability increase. Furthermore, we noticed that we accidentally have included wrong figure in our earlier version of the manuscript, i.e. the one that contains raw datapoints instead of modelled ones. Thus, we have now replaced our old figure with the figure that shows these modelled datapoints (see also line 315). In this figure the slope is somewhat steeper than in the original figure and if we would extrapolate the slope to 0 dissimilar alleles, the viability increase over the full dissimilarity range would be close to 10%. As biological data is often ‘noisy’, we tend to think that this slope may be far from insignificant and may represent a biologically important phenomenon. However, we agree that HLA similarity likely may not lead to complete fertilization failure, but rather can reduce the chances of conception. As reviewer 1 also commented on this, we have now toned down our claims regarding infertility (see abstract and lines 397-399). Finally, in addition to the association between HLA dissimilarity and sperm viability, we also found that male-female interaction (‘compatibility effect’) explained relatively large proportion (up to 32%) of the total variation in measured sperm traits. In fact, the magnitude of this effect was often much higher than the main effect of female and occasionally nearly as large (or even larger) as male effect. Consequently, we believe that genetic compatibility likely play a major role behind our findings. However, it is also likely that HLA-dissimilarity captures only part of the variation in male-female compatibility.